# Artificial and natural selection components reveal the mechanisms of tropical sheep populations against gastrointestinal parasites

Leonardo Sartori Menegatto[1¤]*, Karine Assis Costa[2], Ricardo Dutra do Bem[2], Luara Afonso de Freitas[1], Luiza Vage Coelho Sartori[2], Elisa Peripolli[3], Nedenia Bonvino Stafuzza[2], Claudia Cristina Paro de Paz[4]

1 Department of Genetics, Ribeirão Preto Medical School (FMRP), University of São Paulo (USP), Ribeirão Preto, São Paulo, Brazil, 2 Beef Cattle Research Center, Animal Science Institute (IZ), Sertãozinho, São Paulo, Brazil, 3 Center of Agrarian Sciences (CCA), Federal University of Santa Catarina (UFSC), Florianópolis, Santa Catarina, Brazil, 4 Sustainable Livestock Research Center, Animal Science Institute (IZ), São José do Rio Preto, São Paulo, Brazil

¤ Current address: Department of Genetics, Ribeirão Preto Medical School (FMRP), University of São Paulo (USP), Ribeirão Preto, São Paulo, Brazil.

* leonardomenegatto@gmail.com

## Abstract

The infection by *Haemonchus contortus* is a significant challenge to sheep production in tropical regions, particularly in developing countries. Although several genomic studies have been conducted on this topic, there is still a lack of research combining evolutionary information on resistance and resilience to nematode infection. The aim of this study was to provide evidence of different types of selection and their effects on traits associated with infection levels and animal productivity, using pedigree, phenotypic, and genomic data. It was hypothesized that these patterns would reflect indirect artificial selection and relaxed natural selection. Phenotypic data were collected for Faecal Egg Count (FEC), Eye Color Chart (ECC), Packed Cell Volume (PCV), Total Plasma Protein (TPP), Body Weight (BW), and Body Condition Score (BCS) from 1,283 Santa Inês sheep. A total of 638 animals were genotyped using the Ovine SNP50 BeadChip. After estimating breeding values using the BLUPF90 software, statistical models were employed to assess differences in the intensities of natural and artificial selection and to identify the type of selection acting on each trait, in comparison with classic studies of sexual selection. Selection signatures were investigated using Wright's fixation index, in addition to analyses of runs of homozygosity. The gene content of the identified regions and their associated pathways were examined using the Ensembl BioMart tool and the Panther Classification System, respectively, along with alignments of quantitative trait loci (QTL). BCS was found to be the best indirect trait correlated with parasitological traits, and selection intensity analysis showed that natural selection contributed 76%, compared to 24% from artificial selection. Traits such as BW, PCV, and TPP exhibited directional selection, while FEC and ECC varied according to the challenge level applied. A total of 15 selection

**Data availability statement:** All relevant data are within the manuscript and its Supporting information files.

**Funding:** This work received funding from Fundação de Amparo à Pesquisa do Estado de São Paulo - Brasil FAPESP (Process Number 2016/14.522-7). The funders had no role in study design, data collection and analysis, decision to publish, or preparation of the manuscript.

**Competing interests:** The authors have declared that no competing interests exist.

**Abbreviations:** BCS, Body Condition Score (beef production trait evaluated); BW, Body Weight (beef production trait evaluated); ECC, Eye Color Chart (parasitological trait evaluated); FEC, Faecal Egg Count (parasitological trait evaluated); PCV, Packed Cell Volume (parasitological trait evaluated); TPP, Total Plasma Protein (parasitological trait evaluated)

signatures were identified (11 for natural selection and 4 for artificial selection), with 9 overlapping with islands of homozygosity, encompassing 131 genes and 49 QTL. A critical analysis revealed that both types of selection contribute to the phenomena of resistance and resilience. However, evidence of directional selection, hard sweeps, and functional enrichment of innate immunity was found for artificial selection, while natural selection exhibited evidence of stabilizing selection, soft sweeps, and functional enrichment of adaptive immunity.

## Introduction

Infections by gastrointestinal parasites pose a significant challenge to sheep (*Ovis aries*) production worldwide, with particular concern over hematophagous nematodes in developing countries, particularly *Haemonchus contortus* in tropical regions [1,2]. Genetic breeding using rustic breeds as maternal lines offers great potential for improving resistance to parasites in sheep herds in these countries, with promising results observed in the Santa Inês breed [3–5]. In this context, research combining molecular and phenotypic data has become increasingly important, with an evolutionary approach being one of the most relevant strategies for understanding the genetic history of parasitological traits [6], including conserved genomic regions such as islands of homozygosity and selection signatures [7,8].

Considerable efforts have been made to identify these conserved regions in both natural [9–19] and artificial populations [8–27], using methods such as runs of homozygosity (ROH) [8,19] or selective sweep analysis [8–18,20–27]. These studies have employed various historical perspectives [9–21,27], with selection focused on agronomic traits [8,22,25] or specifically on parasitological traits [23,24,26]. The results of these studies have confirmed the identification of different biological mechanisms that provide resistance to gastrointestinal worms in sheep, including traditional innate and adaptive immune responses, gastrointestinal mucosal protection, and homeostatic biochemical pathways [28,29]. More specifically, Estrada-Reyes et al. [26] identified selection signature outliers among different sheep populations, revealing important insights into resistance to *H. contortus.*

However, it remains unclear how sheep populations have developed evolutionary adaptations to hematophagous gastrointestinal nematodes, particularly considering the varying selection strategies in their adaptive landscape – either to avoid infection (resistance) or to tolerate infection without developing anemia (resilience) [3,30]. Given the absence of breeding programs focused on parasitological traits in tropical regions [31], it is hypothesized that these genomic regions may have been subject to relaxed natural selection or undirected artificial selection, as proposed by Price [32]. Relaxed selection can be understood as weak selection, similar to that which would occur in a wild environment, but to a lesser degree due to issues such as health and nutritional treatment, food supply, conflict management, and shelter. In turn, indirect artificial selection is the directional selection for a trait in a breeding program, but which ends up selecting other trait not initially intended by the breeder athwart genetic draft due to correlation.

This topic intersects with the study of selection components, drawing from the foundational works of Arnold & Wade [33,34] on natural and sexual selection, which expanded on earlier ideas proposed by Fisher [35], Lande [36,37], and Lande & Arnold [38]. While artificial selection is often equated with runaway selection, similar to sexual selection, the key distinction lies in their respective aims. This study proposes to apply the framework of selection components, as suggested by Arnold & Wade [33], to the Santa Inês breed. Additionally, we utilize other established methods in Evolutionary Biology to provide new perspectives on resistance and resilience to gastrointestinal nematodes.

We hypothesize that the lack of systematic artificial selection for parasitological traits would lead to a greater influence of natural selection on the evolution of these traits, resulting in a net reinforcement of selection pressures. Using different infection challenge levels (across five populations) and data on genomics, pedigree, and phenotypic measurements, this study aims to: (1) determine the proportion of natural versus artificial selection on resistance and resilience to gastrointestinal nematodes; (2) assess whether high or low infection challenge levels affect the type of net selection experienced by each population (neutral, directional, stabilizing, or disruptive); and (3) identify putative candidate genes and loci associated with regions potentially under natural or artificial selection, using selection signature methods and combining this with data on islands of homozygosity.

## Results

### General results by farm and trait

We measured two production traits – Body Weight (BW) and Body Condition Score (BCS) — and four parasitological traits — Packed Cell Volume (PCV), Total Plasma Protein (TPP), Eye Color Chart (ECC), and Faecal Egg Count (FEC) – across five Santa Inês breed populations. The means and standard deviations for all parameters analyzed across the five farms are shown in Table 1. Correlations between production traits and parasitological traits are presented in Table 2. The age as fixed effect ranged from zero to 784 days (19.6% 0 days, 13.4% 1–150 days, 13.5% 151–550 days, and 53.5% over 550 days).

Table 1, Pages 6 and 7. Means ± standard deviation of Body Weight (BW), Packed Cell Volume (PCV), Total Plasma Protein (TPP), and Faecal Egg Count (FEC) traits in Santa Inês sheep populations across five farms. The table includes minimum-maximum (min-max) values and the coefficient of variation (CV). Body Condition Score (BCS) and Eye Color Chart (ECC) were not included, as they are non-parametric parameters. The CV was not calculated for PCV and FEC due to the nature of these parameters: the first is highly variable, and the second is discrete. Considering that there was

**Table 1. Measures of central tendency and dispersion of the parasitological parameters evaluated.**

| Trait | Farms | | | | | Min-max | CV |
|---|---|---|---|---|---|---|---|
| | 1 | 2 | 3 | 4 | 5 | | |
| BW (Kg) | 45.17 ± 9.21**b** | 40.79 ± 8.67 | 40.57 ± 12.60 | 57.23 ± 14.39 | 58.63 ± 11.53**a** | 3.00-139.00 | 0.23 |
| PCV (%) | 27 ± 5.90**b** | 27 ± 6.00 | 29 ± 6.40 | 30 ± 7.10 | 30 ± 7.30**a** | 10-46 | -- |
| TPP (g/dL) | 5.93 ± 3.71**b** | 6.03 ± 3.69 | 6.57 ± 4.00 | 6.59 ± 4.18 | 6.90 ± 4.12**a** | 2.00-9.80 | 0.61 |
| FEC (egg/g) | 610 ± 1,260**a** | 578 ± 1,144 | 556 ± 920 | 563 ± 1,330 | 539 ± 682**b** | 0-15,364 | -- |

**Table 2. Correlation between parasitological and production parameters.**

| Trait | BW | BCS | $h^2$ | $\sigma_a^2$ |
|---|---|---|---|---|
| PCV | 0.29 ± 0.07 | 0.53 ± 0.08 | 0.38 ± 0.04 | 4.98 ± 0.74 |
| TPP | 0.44 ± 0.07 | 0.44 ± 0.10 | 0.29 ± 0.03 | 0.08 ± 0.02 |
| ECC | −0.14 ± 0.13 | −0.41 ± 0.13 | 0.20 ± 0.05 | 0.12 ± 0.03 |
| FEC | −0.16 ± 0.12 | −0.26 ± 0.18 | 0.13 ± 0.04 | 0.32 ± 0.05 |

a contrast test between farms 1 and 5 regarding infection level and selection pattern, both were compared using Tukey's test at a 5% significance level after analysis of variance confirmed the difference. Different letters indicate significant differences between sites, indicating that the four traits diverged.

Table 2, Page 7. Pearson correlations ± standard deviations of Packed Cell Volume (PCV), Total Plasma Protein (TPP), Eye Color Chart (ECC), and Faecal Egg Count (FEC) parasitological traits with Body Weight (BW) and Body Condition Score (BCS) production traits in Santa Inês sheep populations. Mean heritabilities ($h^2$) in narrow sense are available to each parasitological trait, even as the additive genetic variance ($\sigma_a^2$).

The farms with the highest (Farm 1) and lowest (Farm 5) challenge levels were located in the same region of São Paulo State (Cravinhos and Pontal counties, respectively). This distribution does not seem to be influenced by climate conditions but is likely the result of differences in animal management practices, sanitary control measures, and the specific infection histories at each location. Consistently, Farm 1 exhibited the highest FEC (610 ± 1,260 eggs/g) and the lowest levels of PCV (0.27 ± 0.59%) and TPP (5.93 ± 3.71 g/dL), while Farm 5 showed the highest levels of PCV (0.30 ± 0.73%) and TPP (6.90 ± 4.12 g/dL), and the lowest FEC (539 ± 682 eggs/g) among all farms analyzed. Furthermore, both BW and BCS were higher on Farm 5 compared to Farm 1 (58.63 ± 11.53 vs. 45.17 ± 9.21 kg, and 2.89 ± 0.95 vs. 2.32 ± 0.87, respectively). The management practices that generated such differential levels of infection are related to the frequency of deworming, general sanitary control, disposal of infected matrices, and level of evaluation.

The correlation patterns did not show variation among farms. BCS exhibited the highest mean correlation with the four parasitological traits, with a marked difference from traits with lower heritability, as previously reported in sheep breeding [24]. TPP showed the highest correlations overall and PCV the highest correlation with BCS, while FEC displayed the lowest correlations in both cases. ECC also showed a moderate correlation with BW, with an almost threefold increase when switching to a production trait (see Table 2).

Despite these results, the contrast between the farm with the highest challenge (1) and the farm with the lowest (5) challenge was detected by a mean comparison test, after analysis of variance that detected a difference at the 5% significance level. The four traits measured in this regard — BW, PCV, TPP, and FEC — showed differences between the two farms (see Table 1).

## Selection components

The combined analysis based on the adaptation of the classic study by Arnold & Wade [34] to estimate the relative contribution of each component to the evolution of parasitological traits revealed 76% natural selection and 24% artificial selection, using BCS as reference. In the original study, the authors developed a model for episodes of directional selection, defining such episodes as segments of the selection gradient in a life cycle, measuring whether the development of certain characters is due to an adaptive advantage or to female preference for males. Interestingly, this result is almost the opposite of what Arnold & Wade [34] found for the Jamaican turquoise (*Anolis grahami*) in Trivers' [39] study on sexual selection, considering that the basal selection in the present study is of the runaway selection type, rather than natural selection.

The original clustering had found 196 susceptible, 288 resistant, and 263 resilient animals. Considering the use of genomic data (493 animals), 157 were grouped as susceptible, 181 as resistant and 155 as resilient. After STRUCTURE analysis, it was detected differences among the three groups ($p_{resistant-suceptible} = 0.062$, $p_{resistant-resilient} = 0.062$, and $p_{resilient-suceptible} = 0.062$), demonstrating that the groups present genetic differences and corroborating the pedigree alignment (Fig 1). The distribution of FEC and PCV by cluster is available in Table 3, considering all the five farms grouped.

Fig 1, Pages 9 and 10. The genetic admixture was analysed by STRUCTURE detection. On the right, there is the hierarchical classification produced by the ggplot2 R package. On the left, the classification is ordered by any cluster genetic admixture.

Fifteen putative selection signatures were identified, corresponding exclusively to either natural or artificial selection (11 and 4, respectively), which is consistent with the proportions of selection components calculated using phenotypic data.

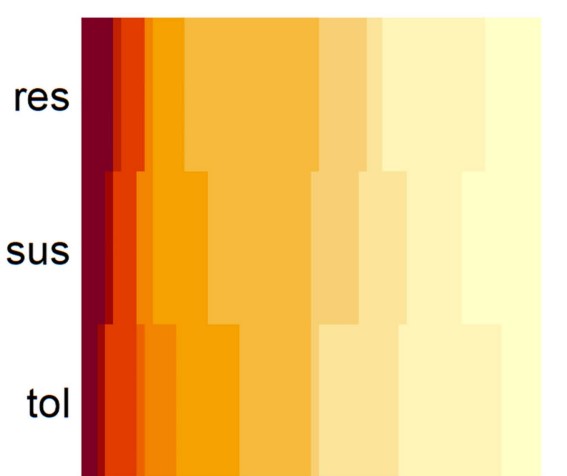 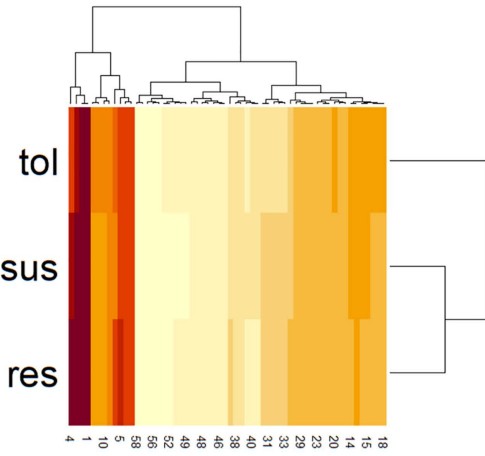

**Fig 1. Heatmap produced from genetic admixture against resistant (*res*), resilient (*tol*) and susceptible (*sus*) groups after a cluster analysis using immune phenotypic data.**

**Table 3. Faecal Egg Count (FEC) and Packed Cell Volume (PCV) averages before (total) and after clustering among resistant, resilient and susceptible groups.**

| Cluster | FEC (egg/g) | PCV (%) |
|---|---|---|
| Total | 829 ± 1,007 | 29 ± 7.0 |
| Resistant | 247 ± 324 | 29 ± 6.3 |
| Resilient | 1723 ± 1,215 | 30 ± 7.2 |
| Susceptible | 2055 ± 1,492 | 26 ± 5.7 |

A total of 681 islands of homozygosity (ROH) were detected, with an average length of 3,346 bp and an average density of 48.09 bp/SNP (region length per marker). Nine selection signatures were found to overlap with ROHs. Six regions were putatively associated with natural selection and three with artificial selection, with two overlapping in the latter group, resulting in a total of eight differential regions. In all, 14 distinct regions were identified, containing protein-coding genes and quantitative trait loci (QTL). Of these, seven regions contained only QTLs, totaling 131 genes and 49 QTLs.

### Selection type

Using the method proposed by Lande & Arnold [38] and the simplified cubic spline method by Schuler [40], we modeled functions for BW in the farms with the highest (Fig 2a) and lowest (Fig 2b) challenge levels (Farms 1 and 5, respectively). The same modeling approach was applied to the four parasitological traits at the highest and lowest challenge levels: ECC (Fig 3a and 3b), FEC (Fig 3c and 3d), PCV (Fig 3e and 3f), and TPP (Fig 3g and 3h). In the first case, graphs of gain with selection were constructed as a function of genotypic variance for each trait. The second case used a non-parametric model in which the construction of the gain function with selection would be carried out by a cubic spline whose existence of control points would be directly proportional to the roughness of the evolutionary possibilities.

Fig 2, **Page 11.** The adaptations were based on (i) the parametric model of Lande & Arnold [38] and (ii) the non-parametric model of Schluter [40], for (a) high and (b) low challenge level farms to gastrointestinal parasites. Panels 2a and 2b represent linear functions, 2d represents a second-degree cubic spline approximation, and 2c represents a third-degree cubic spline approximation.

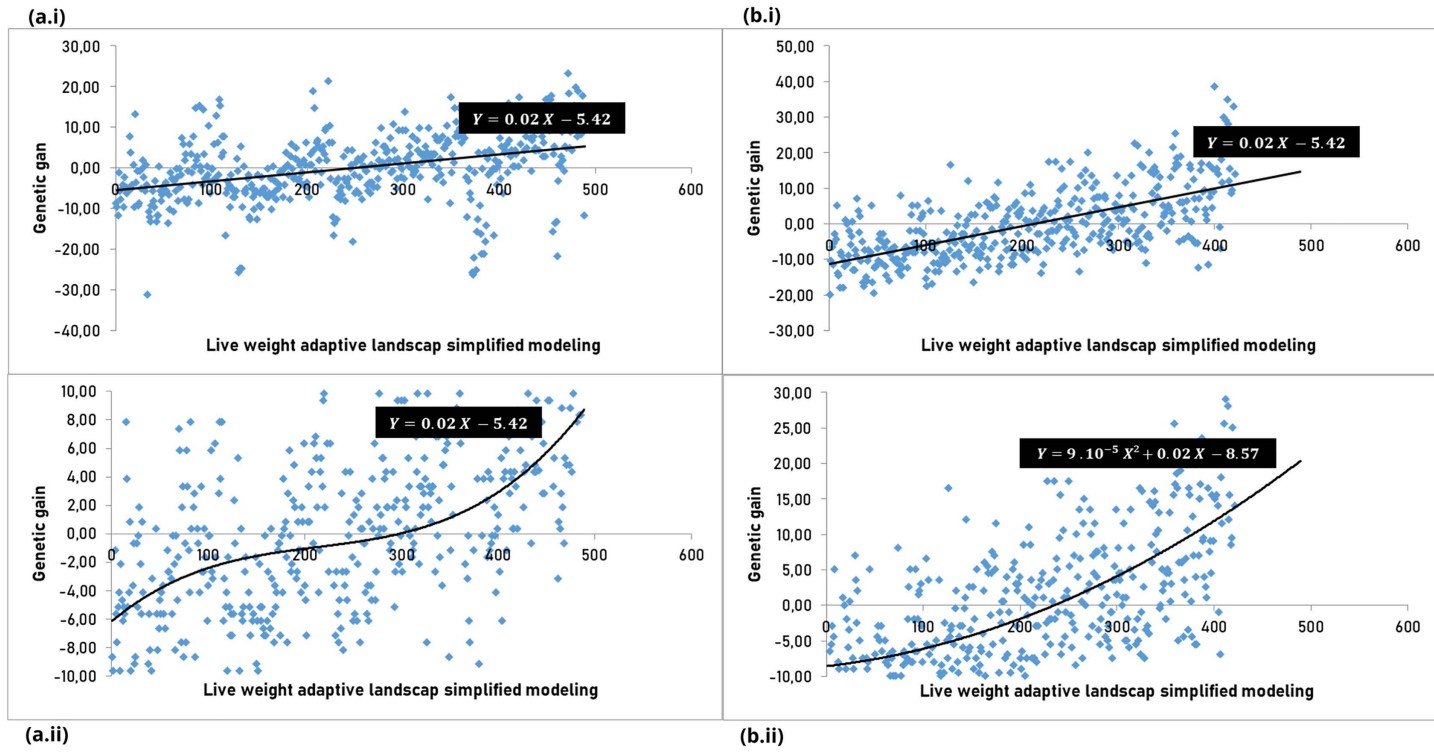

**Fig 2. Data distribution and functions that model the adaptive landscapes of Body Weight (BW) in sheep.**

Fig 3, **Pages 11 and 12.** The parameters analyzed included (a-b) Eye Color Chart (ECC) (in red), (c-d) Fecal Egg Count (FEC) (in green), (e-f) Packed Cell Volume (PCV) (in yellow), and (g-h) Total Plasma Protein (TPP) (in blue). Adaptations based on (i) the parametric model of Lande & Arnold [38] and (ii) the non-parametric model of Schluter [40] for (a-c-e-g) high (light color) and (b-d-f-h) low (dark color) challenge level farms to gastrointestinal parasites were performed. Panel 3c.i represents a constant function (linear function with no significant angular coefficient); 3b.i, 3d.i, 3e.i, 3f.i, and 3h.i represent linear functions; 3a.i, 3f.i, and 3g.i represent second-degree cubic spline approximation functions; 3a.ii, 3f.ii, and 3h.ii represent third-degree cubic spline approximation functions; 3c.ii and 3e.ii represent fourth- degree cubic spline approximation functions; and 3g.ii represents a fifth-degree cubic spline approximation function.

BW was modeled using a linear function in both farms and for both models, with cubic splines not improving data distribution and showing a stronger trend in the low challenge level. A strong directional selection was clearly detected, with the adaptive landscape resembling a slope (Fig 2). A similar result was observed for PCV, where the use of a cubic spline in the high challenge level smoothed the trend (Fig 3e and 3f).

For the other traits analyzed, the low challenge level also showed directional selection, with cubic spline modeling not altering the trend for ECC, while a smoothed trend was observed for FEC and a sharper trend for TPP with this approach. At this challenge level, FEC and ECC followed linear functions with negative angular coefficients, resulting in adaptive landscapes resembling a cliff, consistent with purifying (negative directional) selection (Fig 3b, 3d and 3h).

For the high challenge level, FEC was modeled by a constant function, with no significant contribution from cubic spline modeling, yielding an adaptive landscape resembling a plain, indicating null selection (Fig 3c). ECC was modeled by a hyperbolic function with downward curvature, showing no significant improvement with cubic spline modeling, and the adaptive landscape resembled a mountain, indicating stabilizing selection (Fig 3a). Finally, TPP was modeled by a

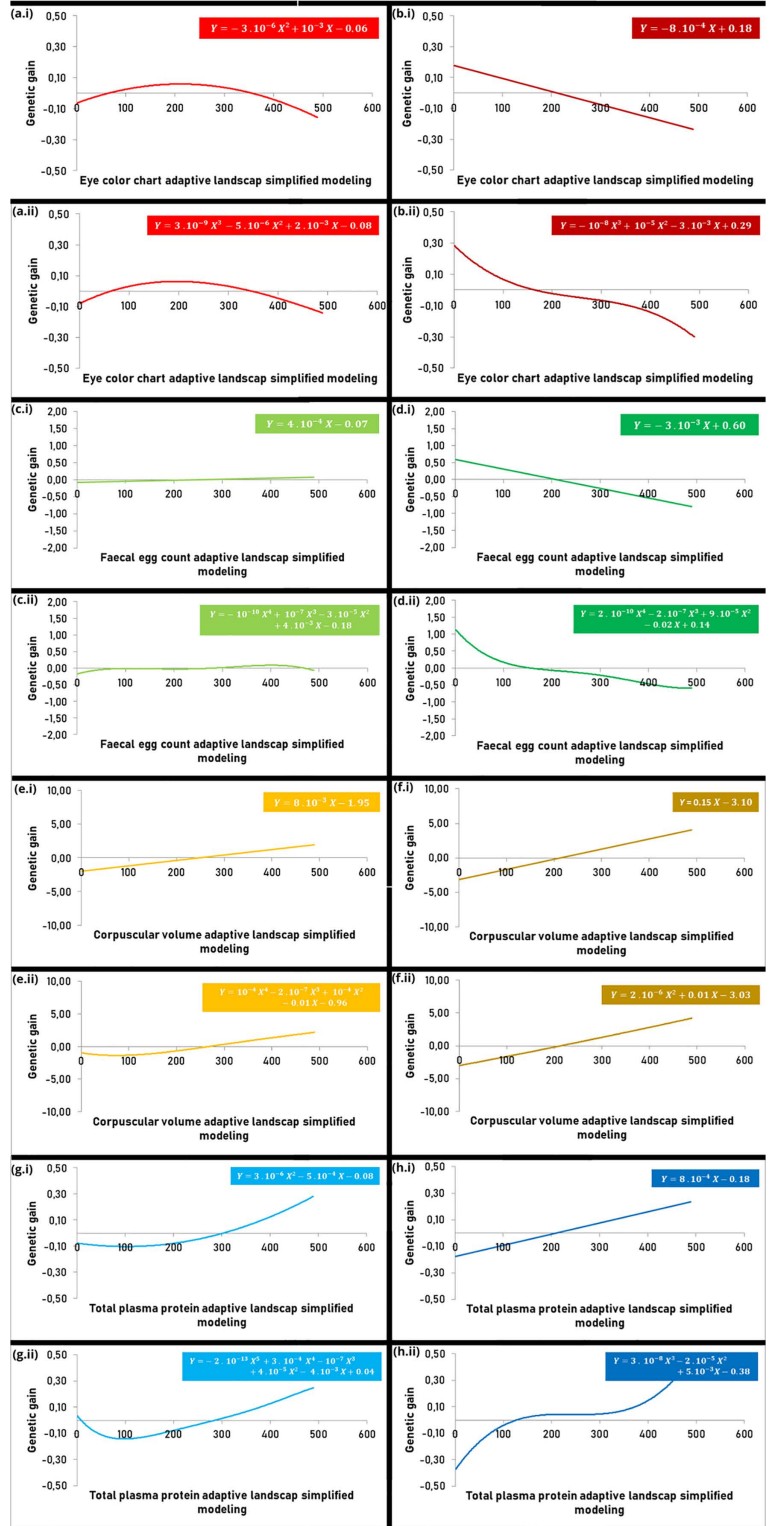

**Fig 3. Functions that model the adaptive landscapes of the parasitological parameters in sheep populations.**

hyperbolic function with upward curvature (Fig 3g.i), with cubic spline modeling smoothing the trend (Fig 3g.ii), yielding an adaptive landscape resembling a valley and indicating disruptive selection.

The genomic regions were also analyzed based on the type of selection they underwent, as described by Estrada-Reyes et al. [26]. For the selection signatures detected in regions not overlapping with islands of homozygosity (ROH), four out of five putative natural selection regions showed stabilizing selection, suggesting a possible relationship between this selection component and this type of selection. In regions overlapping with ROHs, which are favored by directional selection [8], two regions showed stabilizing selection, and four regions showed directional selection. Consistently, all three combined regions putatively under artificial selection exhibited directional selection.

## Annotation of selection regions with QTL data

All gene content and QTL associations per region, along with their genomic coordinates, are detailed in S1 Table. To characterize the functional relevance of these regions, we used the Sheep QTLdb from The Animal QTL Database (https://www.animalgenome.org/cgi-bin/QTLdb/OA) to identify QTLs associated with health, meat and carcass, production, reproduction, and exterior traits.

Health-related traits included blood parameters, congenital defects, disease susceptibility, immune function, mastitis, and parasite resistance. Among the 49 QTLs identified, 27 were exclusive to putative natural selection regions (including six related to health), 17 occurred only in putative artificial selection regions (five related to health), and five were shared between both types of regions (one health-related). No QTLs related to reproduction traits were detected, while the majority (28) were linked to meat and carcass traits. An overlay analysis with runs of homozygosity (ROH) islands revealed significant enrichment of health-related QTLs. Notably, associations with Faecal Egg Count (FECGEN) and Eosinophil Number (CEOSIN) stood out among the 12 health traits evaluated. These traits play a crucial role in immune defense against parasitic infections, as eosinophils, alongside mast cells, are key components of the innate immune response [41]. These QTLs were located within regions under putative natural selection, as were those associated with Hematocrit (HCT) and Changes in Hematocrit (DHCT). In contrast, putative artificial selection regions were enriched for QTLs linked to Creatinine levels (CREAT), Mean Corpuscular Hemoglobin content (MCH), and parasite burden, including *Trichostrongylus* adult and larval count (LATRICH_2)—associated with *T. colubriformis*, a major gastrointestinal nematode in temperate environments—and *Nematodirus* faecal egg count (NFEC).

For meat and carcass traits, QTLs were evenly distributed across regions overlapping and non-overlapping with ROH, as well as between natural and artificial selection regions. However, all fat-related traits were found exclusively within natural selection regions. Similarly, production traits were identified only in these regions and were predominantly linked to growth parameters, such as bone mineral mass and body mass. Among the exterior traits, seven out of eight QTLs were detected only in natural selection regions and were associated with horn type, body conformation, udder traits, and behavior. A single behavior-related QTL was found in artificial selection regions. In contrast, QTLs related to meat color, pH, and sensory characteristics were confined to artificial selection regions, whereas anatomical and fatty acid-related traits were distributed across both groups of regions.

## Gene content analysis

In order to increase the scope of the analysis performed, the ROH and selection signatures results were explored by metabolic pathways in the Panther Classification System (http://www.pantherdb.org/help/PANTHERhelp.jsp) for the cattle genome (*Bos taurus*), considering the absence of the specie as a reference genome (*Ovis aries*); in addition to the assessment of Significant Gene Ontology (GO) terms (biological processes, cellular components, and molecular functions) and KEGG (Kyoto Encyclopedia of Genes and Genomes) pathways for the sheep reference genome itself. Considering the Panther Classification System, from the 131 genes identified in this study, 28 were located in the putative artificial selection regions and 103 in the putative natural selection regions. The pathway analysis revealed six significant

pathways associated with artificial selection and three pathways linked to natural selection (for a schematic representation illustrating the biological pathways related to production and immunity, and their connection to the selective pressures observed in the loci under study, see Fig 4).

Fig 4, **Page 15.** The figure demonstrates a net selection resulting from a smaller proportion of undirected artificial selection and a higher proportion of relaxed natural selection. The pathways identified in each selection are intermingled, contributing to the resistance and resilience of Santa Inês sheep populations against gastrointestinal nematodes, through either innate or adaptive immunity. These pathways also appear to be associated with meat quality.

The artificial selection group showed enrichment in the following pathways: Cholesterol Biosynthesis, β-3-adrenergic Receptor Signaling, Krebs Cycle, Enkephalin Release, Histamine H2 Receptor-Mediated Signaling, and 5HT4 Receptor-Mediated Signaling. In the natural selection group, the significant pathways included Oxidative Stress Response, PI3 Kinase Pathway, and Insulin/IGF Pathway-Protein Kinase B (PKB) Signaling Cascade. The direct functional analysis of the sheep genome identified 187 KEGG terms. Among these, 38 terms (20.3%) were shared between regions under putative natural and artificial selection. The number of KEGG terms associated with natural selection was substantially higher (134 terms; 71.7%) compared to those linked to artificial selection (15 terms; 8.0%) (see S2 Table).

Pathways related to immunological processes were prominent overall, while those associated with muscle deposition were found exclusively within regions under artificial selection. Additionally, a specific island of homozygosity located on chromosome 18, enriched in resistant individuals, was identified and contains the *LRFN5* gene. The β-alanine metabolism pathway was highlighted in the analysis, along with several interrelated pathways such as HIF-1 signaling, cAMP signaling, nicotinate and nicotinamide metabolism, and AMPK signaling. Further significant pathways included cGMP-PKG signaling, aldosterone-regulated sodium reabsorption, mTOR signaling, phospholipase signaling, vitamin D signaling, and renin secretion.

The analysis also detected several neurotransmission-related KEGG pathways, such as glutamatergic, cholinergic, and GABAergic synapses. Additionally, immune-related processes were evident, including T and NK cell signaling, eosinophil recruitment, and various hormone-associated pathways (e.g., cortisol, thyroid hormones, insulin, and adipocytokines). Other relevant metabolic and physiological pathways included oxidative phosphorylation, propanoate metabolism, melanogenesis, and pathways related to environmental adaptation, such as Wnt signaling, circadian entrainment, taste transduction, and arginine and proline metabolism.

Immune system-related pathways such as inflammation, digestive and vasoconstrictive responses, mucin production, coagulation, vasopressin-regulated water reabsorption, relaxin signaling, and platelet activation were also identified. Moreover, the JAK-STAT signaling pathway, leukocyte migration, and neutrophil extracellular trap formation were notably expressed. A total of eighteen infection-related pathways were detected, including those related to African trypanosomiasis and bacterial invasion of epithelial cells.

## Discussion

### Biological pathways related to production and immunity, and their connection to the selective pressures

The identification of the Cholesterol Biosynthesis pathway under artificial selection highlights its critical role in lipid metabolism and its downstream effects on energy production, insulin regulation, and immune function. HDL, a product of this pathway, exhibits multiple protective roles such as antioxidative and anti-inflammatory properties, contributing to innate immunity [42]. Jin et al. [43] linked inhibition of cholesterol biosynthesis to enhanced meat flavor through the production of 1-octen-3-ol, a process mediated by mevalonic acid and acetyl-CoA — an intermediate also involved in the Krebs Cycle, which was the second pathway under selection [44].

The β-3-adrenergic Receptor Signaling pathway, also under artificial selection, is associated with fat mobilization and metabolic regulation. The *ADRB3* gene is known for its higher expression in visceral adipose tissues compared to subcutaneous fat, influencing traits such as growth and carcass quality in sheep [45–47]. Moreover, adrenergic signaling

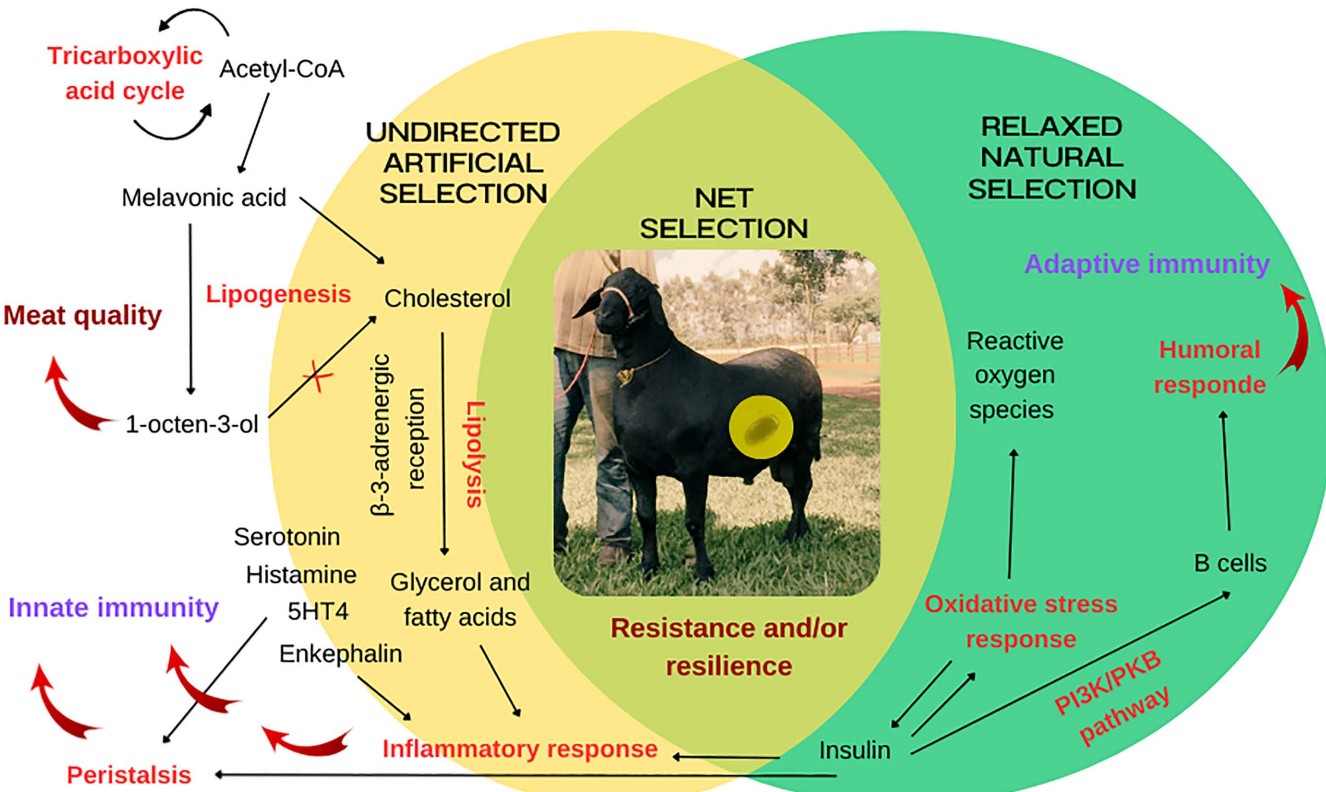

**Fig 4. Illustrative map of the pathways related to beef production and immunity and their interactions under different selective pressures on the studied loci.**

impacts immune function through modulation of cell migration and cytokine production [48]. The Enkephalin Release pathway, another under artificial selection, may suggest selection for immune response traits. Enkephalins, with known anti-parasitic, anti-inflammatory, and analgesic functions, are upregulated in response to parasitic infections and interact with ACTH, a hormone influencing stress and behavioral traits [49–53]. The presence of the Histamine H2 Receptor-Mediated Signaling pathway reflects the potential adaptive relevance of histamine responses. Prior studies in Santa Inês and other breeds identified correlations between histamine levels and immune cell profiles, particularly IgA in the abomasum [41,54–56]. The 5HT4 Receptor-Mediated Signaling pathway, involving serotonin, also plays a role in immune defense by influencing gastrointestinal motility, an important factor in parasite expulsion mechanisms [57–59].

For the natural selection group, the Oxidative Stress Response pathway suggests adaptation to parasitic pressure, as reactive oxygen species (ROS) are implicated in resistance mechanisms to *Haemonchus contortus* and *Trichostrongylus colubriformis* [60–63]. The PI3 Kinase pathway, through genes like *PIK3 CD*, supports immune cell function and regulation, particularly in B cells [64,65]. Its interaction with the Insulin/IGF–PKB Signaling Cascade, where PKB is activated via PI3K in an insulin-dependent manner, links metabolic and immune responses — particularly in apoptosis regulation by lymphocytes [66,67].

The dominance of KEGG terms associated with natural selection suggests that adaptive responses to environmental pressures — especially those related to immunity — may be more extensive than those related to artificial selection, which appears more focused on production traits, such as muscle development. The *LRFN5* gene, found in a homozygous region associated with parasite resistance, is noteworthy due to its role in downregulating macrophage activation and inflammatory responses [68]. Although not included in the GO enrichment analysis, this gene has been identified as under selection in Iranian sheep and is associated with metabolic pathways such as valine, leucine, and isoleucine degradation and retinol metabolism [69,70]. The presence of the β-alanine metabolism pathway, essential for muscle pH regulation [71], along with the AMPK pathway, suggests energy balance and metabolic adaptation. The AMPK pathway's links to calcium metabolism, cell communication, and endocrine regulation [72,73] reveal complex metabolic adjustments possibly selected for in response to production demands.

The identification of multiple signaling cascades — such as mTOR, cGMP-PKG, phospholipase, and vitamin D signaling — highlights their roles in energy sensing, inflammation, and nutrient homeostasis [74–79]. Toll-like receptor (TLR) signaling, observed to be elevated, reflects its central role in early immune responses and its regulation via negative feedback mechanisms [80]. Neurotransmitter pathways (glutamatergic, cholinergic, GABAergic) were also prominent, potentially influencing immune responses through γ-aminobutyric acid (GABA)-mediated modulation of cytokine secretion, immune cell proliferation, and migration [81]. The link between TNF-α and IL-1 — both inflammatory cytokines — has been associated with severe symptoms and impaired protective immunity in sheep infected with *H. contortus* [77,82]. The overrepresentation of CD proteins in T and NK cell signaling supports their known roles in eosinophil recruitment, a key defense mechanism in allergic and helminth infections [83–85]. Additionally, several hormone-associated pathways (e.g., cortisol, thyroid, insulin) were identified, suggesting that both metabolic and endocrine factors contribute to immune and production-related traits.

Metabolic pathways such as propanoate metabolism, oxidative phosphorylation, and melanogenesis further support the involvement of energy metabolism and pigmentation in adaptation [86]. Notably, pathways related to Wnt signaling, circadian rhythms, taste perception, and arginine/proline metabolism may reflect adaptations to environmental changes or introgression events from wild species [87–92]. The detection of infection-related pathways, including JAK-STAT signaling, leukocyte migration, neutrophil trap formation, and those associated with African trypanosomiasis and bacterial infections, confirms the functional importance of the identified genes and pathways in host-pathogen interactions [93].

## Evolutionary implications considering the entire panorama of findings

The first way to categorize selection was discussed in this study, focusing on the indirect action of selection on parasitological traits in sheep, specifically artificial selection. The BCS trait not only stands out as the best selection criterion

due to its correlation with four immune traits, but it is also considered an easy, low-cost tool for detecting variations in fat reserves and muscle volume, making it a viable indirect selection criterion for animals with higher live weight [94,95]. Natural selection, however, is considered to act more directly, albeit in a relaxed manner.

Given that parasitological trait selection in sheep is uncommon in Brazil [31], the threefold greater intensity of natural selection compared to artificial selection is justified. Although stabilizing selection is a notable proxy for parasitological traits [6], when the population is exposed to a lower challenge level (i.e., with more control and likely greater artificial selection), the latter force can modify the intensity and type of selection on these traits, as observed in the directional selection trend when animals were modeled this way. Clearly, at higher levels of challenge, selection is closer to a stabilizing model or soft selective sweeps, and there is also a greater probability of describing a more complex adaptive landscape, where the cubic spline modeling approach proves highly valuable. In the same vein, the method by Foll & Gaggiotti [96] revealed a relationship between stabilizing selection in natural selection *Fst* regions and directional selection in artificial selection *Fst* regions, with deviations likely attributed to the nature of ROH, as described.

For each parasitological trait, TPP showed a high correlation with both BW and BCS. The disruptive selection result, observed at high challenge levels, and its sensitivity to cubic spline adjustments suggest that this trait may sometimes show reinforcement net selection, and other times concurrent net selection between artificial and natural selections (or a more complex adaptive landscape), critically dependent on the challenge level. ECC demonstrated a strong contribution from natural selection through reinforcement net selection, and considering that it is also a simple criterion that could be used to detect anemia [97,98]. Studies showing moderate correlations between ECC detection using the FAMACHA® method and other immunological and production traits in Santa Inês sheep, as reported by Sarmento et al. [99], Oliveira et al. [100], and Berton et al. [101], suggest the potential of this trait as a selection criterion in beef sheep breeding [95]. The high contribution from natural selection was further corroborated by findings of stabilizing selection, particularly at high challenge levels.

FEC and PCV showed evidence of responding to artificial selection with significant reinforcement of natural selection. However, FEC demonstrated lower detection power. Indeed, PCV showed clear signs of directional selection, regardless of the challenge level, although it was influenced by it. In contrast, FEC exhibited directional selection only at low challenge levels, which could be attributed to its low heritability. This conclusion suggests that FEC is a weaker trait for altering the nature of parasite resistance; meaning natural and artificial selection are intermingled in conferring resistance or resilience to gastrointestinal nematodes.

Finally, the QTL and functional analyses helped clarify the selection components involved in resistance and resilience phenomena. Artificial selection showed enrichment for creatinine, which is directly proportional to muscle deposition [102], and for meat quality quantitative traits. The identification of adrenergic molecules, overexpressed in adipose tissue, and cholesterol, where acetyl-CoA participates in the mevalonate acid pathway, reveals an indirect relationship with meat quality. Based on these results, a biological origin for the correlation between BCS and parasitological traits can be hypothesized, explaining the indirect artificial selection discussed. However, further studies are strongly needed to confirm this hypothesis. Other hypotheses in this regard are related to behavioral traits, which were not explored in this study, considering the relationship between enkephalins and adaptation and behavior, as well as the detection of a behavioral quantitative trait for artificial selection, even though two QTL were found exclusively in regions of natural selection.

From an immunological perspective, the pathways associated with artificial selection showed enrichment for the innate immune system in a broad context, such as the cholesterol findings, as well as for inflammatory and analgesic responses, including histamine action and its relationship with mast cells and basophils. The serotonin pathway involved in parasite expulsion plays a joint role in resistance to nematodes, as classified by Benavides et al. [103], who provided evidence of innate immune responses oxygen-linked glycosylation of mucin proteins. This enrichment of innate immunity in artificial selection is corroborated by the other FEC associations with nematode genera, identified exclusively in this cluster.

 

Although anatomy-associated QTLs were shared by both artificial and natural selection, growth and morphological exterior traits were detected only in natural selection regions. Similarly, fatty acid traits were common to both clusters, with reports on cholesterol and β-3-adrenergic pathways in artificial selection, but fat-related traits were exclusively detected in natural selection putative regions. From the perspective of natural selection, the discussion surrounding adipose tissue is more conditioned to a broader view of fat functions, including many important metabolic pathways that enhance fitness. Clearly, themes of strength, stabilizing selection, and natural selection are closely linked.

On the other hand, it can be suggested that specific interactions are uniquely tied to natural selection. Differences in correlations between antibodies may result from the sensitive interaction between immune responses and previous exposure to pathogens [29,104], with antibodies showing considerable individual variation in their development, especially regarding exposure to parasitic nematodes, food availability, and climate conditions [105–107]. In this context, although natural selection acts on a broader set of traits with a more undefined direction, favoring soft sweeps and stabilizing selection, it is precisely this kind of selection that serves as the driving force for antibodies and their specific action against parasites. This strongly justifies the results related to ROS and B cells.

This study adds new insights into understanding resistance and resilience to nematodes from an evolutionary perspective. Body condition score (BCS) emerged as the best trait to use as an indirect selection criterion for parasitological traits in sheep breeding compared to body weight, considering its high correlation with parasitological traits and practicality of evaluation. Natural and artificial selection tend to reinforce each other, particularly in situations where artificial selection is not overly intense, such as in rustic sheep farms in developing countries, with the potential to shift in response to different parasite challenge levels. Indeed, both selection components jointly contribute to both resistance and resilience, although evidence suggests that natural selection is more associated with stabilizing selection, soft sweeps, and adaptive immunity, while artificial selection is more related to directional selection, hard sweeps, and innate immunity.

## Materials and methods

### Dataset and phenotypic collection

The animal study protocol was approved by the Institutional Animal Care and Use Committee of the Animal Science Institute, Nova Odessa, São Paulo, Brazil (Protocol Code CEUA Nº. 267−18, 3 October 2018).

This study used a total of 1,283 Santa Inês sheep of varying ages (3,845 measurements) from five farms located in the states of Paraná and São Paulo (South/Southeast Regions of Brazil). The farms were located in Cravinhos (21°20′25″ S, 47°43′46″ W), Jardinópolis (21°01′04″ S, 47°45′50″ W), Nova Odessa (22°46′39″ S, 47°17′45″ W), Pontal (21°01′21″ S, 48°02′14″ W), and Ventania (24°14′45″ S, 50°14′34″ W). The animals were born between 1999 and 2018, and all the farms are located between 21º and 23º S latitudes, characterized by a humid subtropical climate and tropical climate with a dry season, according to the Köppen classification [108].

The phenotypic traits were collected three to six times at median intervals of 30 days, including body measurements, and samples of feces, blood, and hair. Fecal samples were collected individually and directly using a rectal ampoule and egg counts were performed as described by Gordon & Whitlock [109]. The eggs were analyzed via microscopy [110] to identify proportions of Trichostrongylidae, *Moniezia* sp., *Eimeria* sp., and *Strongyloides* sp., which were used to adjust the Faecal Egg Count (FEC) trait, so that the total available values of this parameter are the result of the sum of of these pathogens. Coprocultures were conducted on pooled samples [111] to identify and quantify the proportion of *Haemonchus* sp., the most significant gastrointestinal nematode in tropical regions, which represented 63% of the total eggs counted, followed by *Trichostrongylus* (24%).

Blood samples were collected by jugular vein puncture into 5 mL vacuum tubes containing 1% EDTA-K3. Total Plasma Protein (TPP) and Packed Cell Volume (PCV) were determined using a microhematocrit centrifugation technique [112]. The Eye Color Chart (ECC) was assessed using the FAMACHA® method, where the ocular conjunctiva was inspected by

two trained evaluators at the time of collection to increase accuracy, using the average of values when they differ by up to one degree between the evaluators, discarding evaluations that differed by a higher degree. The color of the conjunctiva, ranging from robust red to white, was classified on a scale from one to five, with anemia being inversely proportional to pigmentation. The use of the FAMACHA® method is justified as an indicator of anemia, which has a strong positive correlation with gastrointestinal parasitism, particularly *Haemonchus* sp. [97,113].

In addition to these, Body Weight (BW), measured in kilograms (kg), and Body Condition Score (BCS), classified on a scale from one to five based on observation and palpation of the dorso-lumbar and spine regions, were recorded. These traits are important indicators of animal health and are commonly used as selection criteria in sheep breeding [94,114].

Given the non-normal distribution of FEC due to its discrete nature and variation among farms in the dataset, the FEC data were transformed using $\log_{10}$ (n + 25). Pearson correlation [115] was used to assess the relationships between production traits (BW and BCS) and parasitological traits (FEC, PCV, TPP, and ECC), resulting in eight interactions that were tested for statistical modeling accuracy. This analysis also helped assess the direct and indirect selection forces, acknowledging that parasitological traits were not always used as selection criteria, despite their heritability and variability [116], which remains a challenge in Brazil [31,117]. Heritability was estimated using the narrow-sense equation [118].

Cluster analysis was conducted using phenotypic data and pedigree information, as described by Freitas [119]. Pedigree was recorded by herdbooks from 4,821 animals from about seven generations and aligned by kinship matrices. The statistical model considered was:

$$y = X\beta + Z\alpha + W\gamma + \varepsilon$$

which y is the vector of n observations of the response variable(s); $\beta$ is the vector of fixed and covariate effects; $\alpha$ and $\gamma$ are the vectors of the genetic-additive and permanent environment effects; $\varepsilon$ is the vector of random residues; and $X$, $Z$ and $W$ are the incidence matrices for fixed, genetic-additive and permanent environment effects, respectively.

Breeding values were estimated using the BLUPF90 family of programs. Phenotypic traits were estimated using random regression based on Legendre polynomials, considering repeated measurements over time. Covariances for FEC, PCV, and ECC were estimated by Bayesian inference in a single-trait animal model using the THRGIBBS1F9 program (http://nce.ads.uga.edu/wiki/doku.php?id=readme.thrgibbs1) [120], with posterior estimate obtained using POST-GIBBSF90 (http://nce.ads.uga.edu/wiki/doku.php?id=readme.postgibbs) [120,121].

Variance components were estimated using a single-trait animal model that included additive genetic, permanent environmental (fixed effects), and residual effects as random effects. Fixed effects included birth season (1 = animals born from November to April, 2 = animals born from May to October), the season of phenotype collection (idem), birth year, age class (1 = 1–150 days, 2 = 151–550 days, and 3 = more than 550 days), farm (1–5), sex, functional class (e.g., born, weaned lambs, sows in the peripartum period, other sows, and breeders), ECC evaluator, BCS evaluator, and deworming status (categorical) to produce contemporary groups, considering only the significant effects. Additive and residual genetic effects were considered as variable effects for each trait. The final cluster analysis was performed using the Statistica8 program (Statistica 8.0, Statsoft Inc., Oklahoma, USA), with a hierarchical scheme in order to choose the possible number of clusters for the population. Each group was formed from Ward's algorithm [122] and the Euclidean distance was used as a measure of similarity between the animals. Once the number of groups was defined, a non-hierarchical analysis was carried out using the k-means method [123] to detect the genetic profile of the traits in the population. From this clustering, three groups of animals were discriminated against: resistant, resilient (tolerant) and susceptible to verminosis.

## Selection component analysis with phenotypic data

Considering the well-known differences in challenge levels between farms, all traits were compared pairwise within each farm and across farms, with all 1,283 animals included in the study. The farm with the highest challenge level was

identified as the one with the highest FEC and ECC, and the lowest PCV and TPP, while the farm with the lowest challenge level was characterized by the opposite results. It was expected that the challenge level would show an inversely proportional relationship with the mean BW and BCS. The difference between farms was established using a mean comparison test (Tukey), after analysis of variance that detected a difference at a significance level of 5%.

In addition to calculating means, correlations, variation, standard deviation, and heritability ($h^2$) for all traits, we performed selection differential ($\Delta s$) analysis for the parasitological traits using the difference between the base population mean and the mean of the putative selected parents. This analysis was conducted separately for truncated selection based on BW and BCS, using the R software (https://cran.r-project.org/). Finally, the selection gain (sG) was estimated by:

$$sG = \Delta s \cdot h^2$$

The observed selection intensity ($_oi$) was calculated by:

$$_oi = \frac{\Delta s}{s_p}$$

which $s_p$ is the sample phenotypic standard deviation.

The expected measurement as selection intensity ($_ei$) was estimated by the Person's correlations and correlated traits heritability. It was hypothesized that the selection gain in a parasitological trait, resulting from selection based on BW or BCS, represents an indirect component of artificial selection, while any surplus or lack of selection gain reflects the effect of relaxed natural selection. Unlike other studies, in this case, artificial selection was considered the basal model.

The most correlated trait was considered the best estimator to assess the contribution of each selection component. By examining the presence or absence of its influence and combining the observed intensities of the four traits across different farms, maximum likelihood estimation (MLE) was performed using the EMV R package (https://cran.r-project.org/web/packages/EMV/) to detect the proportion of both natural and artificial selection. Arnold & Wade's [33] proposal was used as the basis, considering an analogous selection opportunity (I) as fitness variance over n generations, based on breeding values, on what:

$$I = \sum_{i=1}^{n} \left( \frac{\Delta s_i}{s_{p_i}} \cdot sG_i \right)$$

The contribution of natural and artificial selection components in a net selection was conducted by comparisons among models selection episode models as performed by Lande & Arnold [38] and tested by Arnold & Wade for [33] to wild animals. A direct parallel was the test in the classic study of body size (in mm) of Jamaican lizards by Trivers [39], whose character showed evolution by both natural and sexual selection.

### Selection signature analysis

A total of 638 DNA samples were genotyped using the Ovine SNP50 Genotyping BeadChip (Illumina). It was selected representative animals in each farm by randomized procedure from the pool of 1,283 animals, considering the lived and measured sheep in 2018. Quality control was performed with PreGSF90 [124], considering samples and single nucleotide polymorphisms (SNPs) with call rate < 90%, as well as SNPs located on autosomal chromosomes [8]. After quality control, all samples were subjected to a selection signature analysis, filtering out SNPs with a Minor Allele Frequency (MAF) < 0.05 across all individuals, as well as SNPs that deviated from Hardy-Weinberg Equilibrium ($p < 0.1 \times 10^{-6}$).

From the set of 638 animals genotyped successfully, 634 were gathered after quality control, which were used to selection signatures detections, and of which 493 were properly clustered into the resistant, resilient and susceptible groups.

From 54,241 single nucleotides polymorphisms (SNPs), a total of 42,961 markers remained after quality control to the selection signatures analysis and 47,528 markers to ROH analysis, what represents respectively 79.2% and 87.6% of the total panel and it is consistent with reported by Purfield et al. [8].

To check if the clustering using phenotypic and pedigree information present genetic differences, it was tested the genetic admixture among the three clusters. To detect similarities between groups, the STUCTURE algorithm with struct-2geno function in LEA R package [125] was performed considering p-value = 0.05. Finally, a Heatmap was generated using pedigree information against clustering by ggplot2 R package (https://cran.r-project.org/web/packages) to check the clustering and admixture analysis quality.

Considering the proposed analogy between sexual and artificial selection, the selection signatures were investigated as proposed by Flanagan & Jones [126]. A Maximum Likelihood Estimation (MLE) was performed considering three clusters: certainly selected animals (high BW, BCS, and ECC), possibly selected animals (high in two of the three traits mentioned above), and probably non-selected animals (one or no high trait). An adaptation of the method first implemented by Monnahan et al. [127] was used, where each identified allele was assigned a value of 1, and the others were assigned a value of $10^{-6}$, using a rate of 2. (*L selected – L null*). The artificial selection model considered the certainly selected and possibly selected animals, while the natural selection model considered the possibly selected and probably non-selected animals, with greater differences between the groups, even though natural selection occurs across all groups.

For pairwise contrasts between populations in the clusters, the Wright fixation index (*Fst*) [128] was used, with R programming. The Weir's [129] estimate was calculated by:

$$Fst = \frac{\sum_{i=1}^{n} (p_i - \bar{p})}{n.\bar{p}\,(1 - \bar{p})}$$

which $n$ is the sample size, $p_i$ is the frequency of the *i*-th allele $p$ and $\bar{p}$ is the average of the frequencies of that same allele.

The False Discovery Rate (FDR) method was implemented using the R package *fdrtool* (https://cran.r-project.org/web/packages/fdrtool/index.html) to correct the p- values, and the outliers were compared as differential selection signatures.

As performed in previous studies on sheep [8,19] and goats (*Capra aegagrus hircus*) [130], runs of homozygosity (ROH) analysis were conducted to infer overlaps between selection signatures and islands of homozygosity. The analysis was based on data that underwent only the first quality control step, without the application of Minor Allele Frequency (MAF) filters as suggested previously [128], and without prior linkage disequilibrium control [8,131–135]. ROH detection was performed using PLINK software v.1.09 (https://www.cog-genomics.org/plink/), employing a sliding window approach of 50 SNPs for each animal separately, as performed by Purfield et al. [8] and also suggested by Muchadeyi et al. [63] as the minimal density. The maximum gap adopted was 250 Kb [62,64], and the default trait threshold in the *homozyg.window* function was used [132–135]. Each ROH was categorized based on its physical length, and the mean sum of ROH was calculated for each size category. After generating the files in PLINK, islands of homozygosity were detected, with a minimum of 50% presence in the population, using the *detectRUNS*R package (https://cran.r-project.org/web/packages/detectRUNS/).

The gene content of regions detected by *Fst*, putatively associated with natural selection and artificial selection, was identified using the *biomaRt 2.54.0* R package (https://bioconductor.org/packages/release/bioc/vignettes/biomaRt) from the Ensembl database 110 (http://www.ensembl.org/biomart). In cases where a region detected by *Fst* overlapped with an ROH, the entire content of the ROH was explored. Otherwise, any selective signature within a 250 kb window around the selective sweep was considered. The Ensembl Biomart tool, using the Ensembl Genes 111 database and the *Ovis aries* (Texel) OAR_v3.1 dataset, was used to identify gene content in significant genomic regions. Significant Gene Ontology terms (biological processes, cellular components, and molecular functions) and KEGG pathways were searched using the DAVID v.2023q4 tool [136], with the ovine genome as the background.

Due to the more detailed description of production traits in the bovine genome (*Bos taurus*) and its similarity to *Ovis aries*, an additional analysis was performed by aligning the genes found across this reference genome. All genes identified in the exclusively putative natural selection regions and all genes identified in the exclusively putative artificial selection regions were grouped by pathways using the Panther Classification System (http://www.pantherdb.org/help/PANTHERhelp.jsp). Additionally, previously reported QTLs were retrieved from the Sheep QTLdb, available through The Animal QTL Database (https://www.animalgenome.org/cgi-bin/QTLdb/OA). This database compiles QTLs identified in earlier studies based on associations with phenotypic traits. In the present study, we did not perform new QTL mapping or association analyses. Instead, QTLs were intersected with the genomic regions identified as putatively under natural or artificial selection to explore possible functional enrichments. These QTLs were categorized by trait class as defined in the database, including health, meat and carcass, production, reproduction, and exterior traits (software denomination).

### Selection type analysis

By using phenotypic data, the selection type for the four parasitological traits was estimated, as well as for Body Weight (BW) as a typical production trait. The results were compared between farms with the highest and lowest challenge levels to assess whether the control level affects the selection type, potentially altering the expected selection trends in the natural environment. The methods by Lande & Arnold [38] and Schuler [40] were employed for the analysis. The cubic spline method by Schuler was simplified, and the eye color chart (ECC) method was also considered as an indirect fitness trait.

The Lande & Arnold [38] adaptation analysis modeled the selection types as follows: neutral selection as a constant function, directional selection as a linear function, disruptive selection as a quadratic function with upward concavity, and stabilizing selection as a quadratic function with downward concavity. The functions were constructed using $_{o}sG$ and genetic variance ($s^2g$) to build the functions. The non-parametric adaptation analysis by Schuler [40] was conducted to build adaptive landscapes using cubic splines, with maximum likelihood estimation to calculate the most accurate graph in the original R programming. The use of Schuler 's cubic spline [40] was necessary to measure possible biases in the Lande & Arnold [38] model, since the adaptive landscape may present excessively complex conformations. Such biases are added to the assumptions of normality and to the unforeseen variation that may occur in life stages, years and geographic regions. Furthermore, Mitchell-Olds & Shaw [137] demonstrated that in some situations, such as in a truncated selection whose truncation point differed from the mean, there could be a biased signal that would suggest a quadratic function for a directional selection.

Finally, all differential regions identified by *Fst* were compared as described by Estrada-Reyes et al. [26] using the BayeScan software (http://cmpg.unibe.ch/software/BayeScan/). The method by Foll & Gaggiotti [96] was applied to detect selection types through Bayesian analysis, where logistic regression models the population coefficients decomposed into specific components shared across all loci ($\beta$) and for specific loci shared in all populations ($\alpha$). This method interprets $\alpha < 0$ as indicating stabilizing selection, $\alpha > 0$ as suggesting directional or diversifying (including disruptive) selection, and $\alpha = 0$ as indicating no selection. It employs a Dirichlet distribution in a Markov chain Monte Carlo (MCMC) method, integrating the effects of each population and each locus for each *Fst* value.

### Supporting information

**S1 Table. Chromosome regions detected by selection signatures using Wright's fixation index (*Fst*), their putative selection components, and information on overlaps with runs of homozygosity (ROH).** The gene content and quantitative trait loci (QTL) associations are described for each genomic region, with QTLs classified as either health-associated traits (Health associated QTL) or other types according to the Animal QTL Database (Other associated QTL). (DOCX)

**S2 Table. KEGG (Kyoto Encyclopedia of Genes and Genomes) pathways identified by DAVID in putative artificial selection genomic regions, natural selection genomic regions, and their overlapping components.**
(DOCX)

**S1 File. Runs of homozygosity of all animals studied and functional analysis.**
(XLSX)

**S2 File. Runs of homozygosity of all animals studied by position.**
(TXT)

**S3 File. Runs of homozygosity of all animals studied by chromosome.**
(TXT)

**S4 File. Runs of homozygosity script.**
(TXT)

**S5 File. Genotyping after quality control and use of the BLUPf90 family program.**
(TXT)

**S6 File. Measurements of samples collected from all animals studied.**
(XLSX)

## Author contributions

**Conceptualization:** Leonardo Sartori Menegatto.

**Data curation:** Leonardo Sartori Menegatto, Karine Assis Costa, Ricardo Dutra Do Bem, Luara Afonso De Freitas, Luiza Vage Coelho Sartori, Nedenia Bonvino Stafuzza.

**Formal analysis:** Leonardo Sartori Menegatto, Nedenia Bonvino Stafuzza.

**Funding acquisition:** Claudia Cristina Paro De Paz.

**Investigation:** Leonardo Sartori Menegatto, Luiza Vage Coelho Sartori.

**Methodology:** Leonardo Sartori Menegatto, Nedenia Bonvino Stafuzza.

**Project administration:** Claudia Cristina Paro De Paz.

**Resources:** Leonardo Sartori Menegatto, Nedenia Bonvino Stafuzza, Claudia Cristina Paro De Paz.

**Software:** Leonardo Sartori Menegatto, Luara Afonso De Freitas, Elisa Peripolli, Nedenia Bonvino Stafuzza.

**Supervision:** Elisa Peripolli, Nedenia Bonvino Stafuzza.

**Validation:** Karine Assis Costa, Ricardo Dutra Do Bem, Elisa Peripolli, Nedenia Bonvino Stafuzza, Claudia Cristina Paro De Paz.

**Visualization:** Leonardo Sartori Menegatto, Luiza Vage Coelho Sartori.

**Writing – original draft:** Leonardo Sartori Menegatto.

**Writing – review & editing:** Karine Assis Costa, Luara Afonso De Freitas, Nedenia Bonvino Stafuzza.

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
