## [Decision Letter · Decision Letter 0]

11 Feb 2025

Dear Dr. Menegatto,

Thank you for submitting your manuscript to PLOS ONE. After careful consideration, we feel that it has merit but does not fully meet PLOS ONE’s publication criteria as it currently stands. Therefore, we invite you to submit a revised version of the manuscript that addresses the points raised during the review process.

We look forward to receiving your revised manuscript.

Kind regards,

Jean-Christophe Bambou

Academic Editor

PLOS ONE

Journal requirements:   When submitting your revision, we need you to address these additional requirements. 1. Please ensure that your manuscript meets PLOS ONE's style requirements, including those for file naming. The PLOS ONE style templates can be found at https://journals.plos.org/plosone/s/file?id=wjVg/PLOSOne_formatting_sample_main_body.pdf and https://journals.plos.org/plosone/s/file?id=ba62/PLOSOne_formatting_sample_title_authors_affiliations.pdf. 2. To comply with PLOS ONE submissions requirements, in your Methods section, please provide additional information regarding the experiments involving animals and ensure you have included details on (1) methods of anesthesia and/or analgesia, and (3) efforts to alleviate suffering. 3. Thank you for stating the following financial disclosure:  [This work received funding from Fundação de Amparo à Pesquisa do Estado de São Paulo - Brasil FAPESP (Process Number 2016/14.522-7).].  Please state what role the funders took in the study.  If the funders had no role, please state: ""The funders had no role in study design, data collection and analysis, decision to publish, or preparation of the manuscript."" If this statement is not correct you must amend it as needed. Please include this amended Role of Funder statement in your cover letter; we will change the online submission form on your behalf.

Reviewers' comments:

Reviewer's Responses to Questions

**Comments to the Author**

1. Is the manuscript technically sound, and do the data support the conclusions?

Reviewer #1: Partly

Reviewer #2: Partly

2. Has the statistical analysis been performed appropriately and rigorously?

Reviewer #1: No

Reviewer #2: I Don't Know

3. Have the authors made all data underlying the findings in their manuscript fully available?

Reviewer #1: Yes

Reviewer #2: Yes

4. Is the manuscript presented in an intelligible fashion and written in standard English?

Reviewer #1: No

Reviewer #2: Yes

Reviewer #1: This is an interesting manuscript that contains a lot of useful information. The sample size is reasonably large and comprehensively analysed. The results on the means, standard deviations and correlations are clear and well presented. The quantitative genetic analyses seem appropriate but it would be useful to present the heritabilities in a table. The QTL data are useful but not very well presented. It is not clear how the QTL were identified. Simply stating 'Of the 49 QTL identified ...' lacks sufficient detail on how they were identified. Similar 'Overlay analysis ...' is unclear without a description of what exactly was done. This section needs to be rewritten to include precise descriptions of what exactly was done.

The main concern is the attempt to quantify natural and artificial selection. The methodology is not clear. Simply citing references and then saying we did it differently is unhelpful. The assumptions and hypotheses are unconvincing. The usual assumption is that there will be genetic correlations among the various traits. The sample size is insufficient to provide precise estimates of genetic correlations but estimates have been provided for other breeds. These genetic correlations could account for the selective improvements observed. Undoubtedly, there is potential for both natural and artificial selection for parasite resistance but trying to quantify their relative contribution with unclear methodology is unconvincing.

I did not see Table S1.

There are problems with the way the information is presented starting with the titles of the authors. eg Postdoctoral Professor and postdoctora.

Should 'Cody condition score' in the abstract be 'Body condition score'?

Page 4 Line 12 'genomic regions' better as 'conserved regions'.

Page 5 Line 5 'relaxed natural selection' might be clearer as 'weak natural selection'. It is not clear how artificial selection can be undirected.

Page 6 line 4. PCV, TTP, ECC and FEC are not immunological traits. Perhaps parasitological or pathophysiological?

Page 6 The legend for Fig 1. is partly repeated in the M & M. The previous study should be cited.

Page 6 Line 23. Do not say that the table includes the CV for all data when it doesn't.

Page 8 'Production trait' not 'productive trait'; this has another meaning.

Page 8 line 13 The 'combined analysis' needs more explanation.

Page 9 line 4. How can genomic regions contain health traits? Please rephrase.

Page 9 Line 8 'modelled functions' needs elaboration.

Page 12 Line 20 'pathways to artificial selection cluster' to 'pathways in the artificial ...'.

Page 13 Line 9 onwards. The Resistant Th2 and susceptible Th1 pathways use fatty acid and carbohydrate catabolism to provide energy.

The discussion on pathways identified by KEGG is long and belongs in the discussion. The attempt to link all the pathways to immune mechanisms is unconvincing.

Page 20 Line 12. O-linked glycosylation of mucin proteins is clearer.

Page 21 Line 10. Perhaps indicate why BCS is preferred to body weight as a marker trait.

Page 22 Line 19 Not clear how or why FEC was adjusted.

Page 22 Line 20 Coprocultures to identify and count nematodes?

Page 28 Line 3. How was the gene content estimated by Fst?

Reviewer #2: The study ‘Artificial and natural selection components reveal the mechanisms of tropical sheep populations against gastrointestinal parasites’ is based on the phenotyping of resistance to Haemonchus contortus in a population of 1283 Santa Inês sheep distributed across 5 farms in the south and southeast regions of Brazil. Research combining molecular (Ovine SNP50 Genotyping BeadChip Illumina) and repeated phenotypic data (BW, BCS, FEC, PCV, ECC, TTP) is conducted, with an evolutionary approach being a relevant strategy for understanding the genetic history of immunological traits, including conserved genomic regions such as islands of homozygosity and selection signatures : breeding values (BLUPF90 software family), intensity of natural and artificial selection (statistics models), selection signatures (Wright's fixation index, runs of homozygosity), genes and associated pathways (Ensembl BioMart tool and the Panther Classification System, QTL alignments), selection type analysis (Lande & Arnold and Schuler methods, Bayescan software). The variable infestation levels between these 5 populations are interpreted as a gradient of natural selection pressures.

The manuscript concludes that both types of selection contribute to the phenomena of resistance and resilience. However, evidence of directional selection, hard sweeps, and functional enrichment of innate immunity was found for artificial selection, while natural selection exhibited evidence of stabilizing selection, soft sweeps, and functional enrichment of adaptive immunity.

This is an impressive work in terms of the quantity of animals phenotyped (but over 20 years) and of results produced, but not detailed enough in each of the methods. The phenotyping protocol, the phenotypes recorded, the animals genotyped and the number of animals are not clearly understood.

- 1,283 sheep were phenotyped 3 to 6 times at a median interval of one month, for a total of 2,241 records!

- Is the pedigree recorded or reconstructed by the genomic matrix?

- Infestation is a count corrected for the estimated rate of H. contortus by coproculture. Why was the multi-species infection rate not considered? Is the correction individual or based on a pool of faeces? Specify the proportion of H.contortus on average (plus minimum-maximum). Does this proportion have an impact on the QTL identified?

- Table 1 add numbers

• PCV is not expressed in the correct unit, in fact in %.

• Infection levels are surprisingly low and very comparable, or unit error?

- Specify the rearing practices that generate these differences in infection levels between farms 1 and 5.

- Table 2 specify pearson correlation

- Is the ECC score a compromise between the 2 evaluators or the average of the 2 evaluators' scores?

- The distribution of observations in the different categories of fixed effects remains an enigma. How does it vary (min-max)?

- The effects of age class and functionnal class are partially confused. They need to be combined.

- The rules for choosing genotyped animals are not specified. The genetic structure of the genomic data is essential for assessing the relevance of the results.

All in all, I recommend major revision to make the attractive conclusions more credible.

**Do you want your identity to be public for this peer review?** For information about this choice, including consent withdrawal, please see our Privacy Policy

Reviewer #1: **Yes:**  Michael Stear

Reviewer #2: No

---

## [Author Response · Author response to Decision Letter 1]

17 Apr 2025

REVIEW COMMENTS TO THE AUTHOR

# REVIEWER 1

This is an interesting manuscript that contains a lot of useful information. The sample size is reasonably large and comprehensively analysed. The results on the means, standard deviations and correlations are clear and well presented. The quantitative genetic analyses seem appropriate but it would be useful to present the heritabilities in a table. The QTL data are useful but not very well presented. It is not clear how the QTL were identified. Simply stating 'Of the 49 QTL identified ...' lacks sufficient detail on how they were identified. Similar 'Overlay analysis ...' is unclear without a description of what exactly was done. This section needs to be rewritten to include precise descriptions of what exactly was done.

The mean heritabilities were inserted in Table 2. The texts in the results and methodology were adjusted, but we do not know if they were sufficiently explained in the way the reviewer intended. The QTLs were not identified in an original way, but compared with a database. If this alternative is considered uninformative, we suggest a new review so that we can suppress the results of quantitative trait loci.

The main concern is the attempt to quantify natural and artificial selection. The methodology is not clear. Simply citing references and then saying we did it differently is unhelpful. The assumptions and hypotheses are unconvincing. The usual assumption is that there will be genetic correlations among the various traits. The sample size is insufficient to provide precise estimates of genetic correlations but estimates have been provided for other breeds. These genetic correlations could account for the selective improvements observed. Undoubtedly, there is potential for both natural and artificial selection for parasite resistance but trying to quantify their relative contribution with unclear methodology is unconvincing.

The methodology was better detailed. Regarding the contribution of each selection component (natural or artificial selection), a parallel was drawn with classic studies involving sexual selection in the 1980s.

I did not see Table S1.

We do not know why it was not available, but in the new submitted file it is possible to see.

There are problems with the way the information is presented starting with the titles of the authors. eg Postdoctoral Professor and postdoctora.

We seek to adapt.

Should 'Cody condition score' in the abstract be 'Body condition score'?

Yes. It was an unseen typo.

Page 4 Line 12 'genomic regions' better as 'conserved regions'.

Changed in text.

Page 5 Line 5 'relaxed natural selection' might be clearer as 'weak natural selection'. It is not clear how artificial selection can be undirected.

The text has been better clarified. We understand that selection can be weak or strong depending on the intensity of selection practiced, and this can vary not only by the degree of directional selection, but also by the correlation with other traits. Thus, relaxed natural selection is not only weak selection, but it is especially weak selection that used to be strong in the wild but is no longer so due to issues such as health care, diet, and shelter. In turn, artificial selection can be indirect when the trait has a high correlation with another trait, which is selected.

Thus, since there are no breeding programs in sheep for resistance to worms, such resistance is the product of either natural selection that was relaxed in a captive environment (weak selection that used to be strong) or selection for another trait that led to resistance by draft (indirect artificial selection).

Page 6 line 4. PCV, TTP, ECC and FEC are not immunological traits. Perhaps parasitological or pathophysiological?

Changed in text.

Page 6 The legend for Fig 1. is partly repeated in the M & M. The previous study should be cited.

Changed in text.

Page 6 Line 23. Do not say that the table includes the CV for all data when it doesn't.

Changed in text.

Page 8 'Production trait' not 'productive trait'; this has another meaning.

Changed in text.

Page 8 line 13 The 'combined analysis' needs more explanation.

Changed in text.

Page 9 line 4. How can genomic regions contain health traits? Please rephrase.

The nomenclature was not given by us, but by the program used as reference. We understand the disagreement and, as previously discussed, this issue can be readjusted.

Page 9 Line 8 'modelled functions' needs elaboration.

Changed in text.

Page 12 Line 20 'pathways to artificial selection cluster' to 'pathways in the artificial ...'.

Changed in text.

Page 13 Line 9 onwards. The Resistant Th2 and susceptible Th1 pathways use fatty acid and carbohydrate catabolism to provide energy.

The discussion on pathways identified by KEGG is long and belongs in the discussion. The attempt to link all the pathways to immune mechanisms is unconvincing.

Changed in text. The option to describe it this way was used to make the discussion more fluid. If it is still long, inadequate and/or unconvincing, we can make a new change, including restricting the scope of the results.

Page 20 Line 12. O-linked glycosylation of mucin proteins is clearer.

Changed in text.

Page 21 Line 10. Perhaps indicate why BCS is preferred to body weight as a marker trait.

Changed in text

Page 22 Line 19 Not clear how or why FEC was adjusted.

Changed in text.

Page 22 Line 20 Coprocultures to identify and count nematodes?

Coproculture was used to estimate the proportion of the nematode species. We changed the text to make it clearer.

Page 28 Line 3. How was the gene content estimated by Fst?

Once the regions were estimated by Fst, the gene content of these regions was investigated, with differences in the analysis windows due to overlap or not with a ROH. Please indicate if this is still not clear.

# REVIEWER 2

Reviewer #2: The study ‘Artificial and natural selection components reveal the mechanisms of tropical sheep populations against gastrointestinal parasites’ is based on the phenotyping of resistance to Haemonchus contortus in a population of 1283 Santa Inês sheep distributed across 5 farms in the south and southeast regions of Brazil. Research combining molecular (Ovine SNP50 Genotyping BeadChip Illumina) and repeated phenotypic data (BW, BCS, FEC, PCV, ECC, TTP) is conducted, with an evolutionary approach being a relevant strategy for understanding the genetic history of immunological traits, including conserved genomic regions such as islands of homozygosity and selection signatures : breeding values (BLUPF90 software family), intensity of natural and artificial selection (statistics models), selection signatures (Wright's fixation index, runs of homozygosity), genes and associated pathways (Ensembl BioMart tool and the Panther Classification System, QTL alignments), selection type analysis (Lande & Arnold and Schuler methods, Bayescan software). The variable infestation levels between these 5 populations are interpreted as a gradient of natural selection pressures.

The manuscript concludes that both types of selection contribute to the phenomena of resistance and resilience. However, evidence of directional selection, hard sweeps, and functional enrichment of innate immunity was found for artificial selection, while natural selection exhibited evidence of stabilizing selection, soft sweeps, and functional enrichment of adaptive immunity.

The overview of the article was the one intended by the authors.

This is an impressive work in terms of the quantity of animals phenotyped (but over 20 years) and of results produced, but not detailed enough in each of the methods. The phenotyping protocol, the phenotypes recorded, the animals genotyped and the number of animals are not clearly understood.

The authors understand the critical review and have tried to make the methodology clearer. We expect additional comments, if necessary, so that a new review can be carried out if the explanation is still considered insufficient.

- 1,283 sheep were phenotyped 3 to 6 times at a median interval of one month, for a total of 2,241 records!

In fact, there was a problem in counting records and this was corrected in the text (3,845 measurements). There were few animals with more than three measurements and the long study time is due to the differences between populations and the generations of animals in some of these populations.

- Is the pedigree recorded or reconstructed by the genomic matrix?

The pedigree was recorded by herdbooks and then aligned by the matrix. Information was added to the text.

- Infestation is a count corrected for the estimated rate of H. contortus by coproculture. Why was the multi-species infection rate not considered? Is the correction individual or based on a pool of faeces? Specify the proportion of H.contortus on average (plus minimum-maximum). Does this proportion have an impact on the QTL identified?

QTLs were not originally identified, but rather regions whose gene content was being explored by selection signature were investigated for QTLs by previously available studies. If such an approach seems unconvincing, we suggest suppressing the information related to QTLs.

The approach of using the Haemonchus sp. proportion is related to the history of damage in the tropical herds analyzed, however other genera were identified, such as Trichostrongylus sp., with the proportion added in the text.

- Table 1 add numbers

• PCV is not expressed in the correct unit, in fact in %.

Changed in text.

• Infection levels are surprisingly low and very comparable, or unit error?

Infections appear low because these are average data, but there is great variability between farms. There are populations that have shown high levels of deworming, although this issue was considered in the modeling.

- Specify the rearing practices that generate these differences in infection levels between farms 1 and 5.

Changed in text.

- Table 2 specify pearson correlation

Added information in the methodology.

- Is the ECC score a compromise between the 2 evaluators or the average of the 2 evaluators' scores?

Added explanation in text. We used the average of values when they differ by up to one degree between the evaluators, discarding evaluations that differed by a higher degree

- The distribution of observations in the different categories of fixed effects remains an enigma. How does it vary (min-max)?

We added information in the text.

- The effects of age class and functionnal class are partially confused. They need to be combined.

The model was added to the text. In fact, these effects were combined. We agree that it was not clear.

- The rules for choosing genotyped animals are not specified. The genetic structure of the genomic data is essential for assessing the relevance of the results.

An analysis with STRUCTURE had been performed in the larger project and was missing from the manuscript. The issue was raised. Information regarding the sampling of animals in terms of grouping clustering was also added.

All in all, I recommend major revision to make the attractive conclusions more credible.

Considering that not all issues may be resolved, given our doubts and comments, we remain open to further review if necessary.

---

## [Decision Letter · Decision Letter 1]

27 Jun 2025

Artificial and natural selection components reveal the mechanisms of tropical sheep populations against gastrointestinal parasites|

PLOS ONE

Dear Dr. Menegatto,

Thank you for submitting your manuscript to PLOS ONE. After careful consideration, we feel that it has merit but does not fully meet PLOS ONE’s publication criteria as it currently stands. Therefore, we invite you to submit a revised version of the manuscript that addresses the points raised during the review process.

We look forward to receiving your revised manuscript.

Kind regards,

Jean-Christophe Bambou

Academic Editor

PLOS ONE

Reviewers' comments:

Reviewer's Responses to Questions

**Comments to the Author**

Reviewer #3: (No Response)

2. Is the manuscript technically sound, and do the data support the conclusions?

Reviewer #3: Yes

3. Has the statistical analysis been performed appropriately and rigorously?

Reviewer #3: Yes

4. Have the authors made all data underlying the findings in their manuscript fully available?

Reviewer #3: Yes

5. Is the manuscript presented in an intelligible fashion and written in standard English?

Reviewer #3: Yes

Reviewer #3: The manuscript titled “Artificial and natural selection components reveal the mechanisms of tropical sheep populations against gastrointestinal parasites” is an interesting one. The study used 1283 Santa Inês sheep distributed across 5 farms in the south and southeast regions of Brazil to explore artificial and natural selection components of resistance to gastrointestinal nematode. The study combining phenotypic, genetic and molecular data. They used different methodology including conserved genomic regions, breeding values, selection intensity, selection signatures, genes and associated pathways, selection type analysis. They found that BCS is the best indirect trait correlated with parasitological traits. The natural selection contributed 76%, compared to 24% from artificial selection. A total of 15 selection signatures were identified (11 for natural selection and 4 for artificial selection), with 9 overlapping with islands of homozygosity, encompassing 131 genes and 49 QTL. They concluded that natural selection is more associated with stabilizing selection, soft sweeps, and adaptive immunity, while artificial selection is more related to directional selection, hard sweeps, and innate immunity.

This study provides new insights into understanding resistance and resilience to gastrointestinal nematodes. The genetic analyses appear appropriate. However, some sections lack clarity or sufficient information and require revision. Particular attention should be paid to the measurement and transformation methods used for FEC.

General comments:

The reported infection levels appear unrealistically low and inconsistent. It seems implausible to accurately determine infection at such levels. Although the methodology states that FEC values were log₁₀-transformed, this is not reflected in the table, where values are still presented as eggs per gram. Furthermore, even if the data were log₁₀-transformed, average values such as 11 and 14 seem unusually high, suggesting a possible issue with either the examination or the transformation process. It is recommended to verify and, if necessary, repeat any analyses involving these values.

The QTLs and genes discussed in the study appear to have been conducted from scan genomic regions identified by selection signature; however, this is not clearly explained in the methodology or results sections. Additionally, using a heading such as 'Quantitative Trait Loci Detection' in the results may be misleading, as the analysis does not involve a direct association with measured phenotypic traits.

Many fixed effects were included in the model, but there is no indication of which had a significant impact.

Some sections of the results are more likely to be considered as discussion which need to be revised and moved to the discussion section.

Specific comments:

Introduction:

Page 4 line 4: “Haemonchus contortus Rudolph” what is Rudolph? It seems to be an author name

Results:

Throughout the results section, there is a recurring error in the abbreviation of Total Plasma Protein, which is incorrectly written as TTP. The correct abbreviation should be TPP

Page 8 line 17-18: “TTP and PCV showed the highest correlations overall, and using BCS, respectively, while FEC displayed the lowest correlations in both cases.” Something missing in the sentence and TTP should be TPP

Materials and Methods:

Page 24 line 15: Details on the number of pedigree records and the number of generations should be provided.

Page 24 line 23: It is unclear what permanent environmental effects were tested in the model.

Page 25 line 7-12: The number of levels or classes for each fixed effect should be clearly specified.

Page 25 line 9: “age class (1 = 0 days, 2 = 1 to 150 days, 3 = 151 to 550 days, 4 = more than 550 days)” It is illogical to use a class for age zero days. How were measurements taken at zero age.

Tables and figures:

The quality of the figures is poor and they need to be reproduced in higher resolution.

Table 1: The numbers for PCV are not presented as %

Table 1 and 2: Total Plasma Protein (TTP) should be Total Plasma Protein (TPP) in the header and inside tables

Table 2: I suggest to include genetic correlation

Table 3: “CV%” should be PCV %

Table 3: Standard deviations or standard errors should be added to the reported averages. Additionally, an ANOVA analysis comparing group means is missing and should be conducted to determine if the differences between groups are statistically significant.

Table S2 is not included and the link produce Table S1

**Do you want your identity to be public for this peer review?** For information about this choice, including consent withdrawal, please see our Privacy Policy

Reviewer #3: **Yes:**  Hadeer M. Aboshady

---

## [Author Response · Author response to Decision Letter 2]

27 Aug 2025

# GENERAL COMMENTS

R: The reported infection levels appear unrealistically low and inconsistent. It seems implausible to accurately determine infection at such levels. Although the methodology states that FEC values were log₁₀-transformed, this is not reflected in the table, where values are still presented as eggs per gram. Furthermore, even if the data were log₁₀-transformed, average values such as 11 and 14 seem unusually high, suggesting a possible issue with either the examination or the transformation process. It is recommended to verify and, if necessary, repeat any analyses involving these values.

A: In fact, the data as presented is incorrect. For presentational reasons, we chose to preserve the raw data in the table, although the transformed data were used—correctly—in the analyses.

In reality, egg counts were performed on one-hundred-gram samples, and the data were later adjusted to one gram. In this process, a simple oversight led to a magnitude error. These are not low infections, even though infections varied significantly within farms and especially between farms. We apologize for the error.

R: The QTLs and genes discussed in the study appear to have been conducted from scan genomic regions identified by selection signature; however, this is not clearly explained in the methodology or results sections. Additionally, using a heading such as 'Quantitative Trait Loci Detection' in the results may be misleading, as the analysis does not involve a direct association with measured phenotypic traits.

A: Thank you for your insightful comment. We agree that the methodological explanation regarding the origin of QTL and gene annotations in the identified regions was not sufficiently clear. We have revised the Results section to clarify that the QTL data were retrieved from the Sheep QTLdb and intersected with the genomic regions identified through selection signature analyses. Additionally, we have changed the section heading from “Quantitative Trait Loci Detection” to “Annotation of selection regions with QTL data” to more accurately reflect the nature of the analysis, which did not involve direct phenotype-genotype associations.

R: Many fixed effects were included in the model, but there is no indication of which had a significant impact.

A: Mention of the contemporary group as to significant effects was added.

R: Some sections of the results are more likely to be considered as discussion which need to be revised and moved to the discussion section.

A: These topics were rewritten and the Discussion was subdivided to mark the discussion of these gene content results and the subsequent general discussion. We also merged the gene content topics into a single topic in the Results.

# SPECIFIC COMMENTS

Introduction:

R: Page 4 line 4: “Haemonchus contortus Rudolph” what is Rudolph? It seems to be an author name

A: Rudolph is indeed the author's name. We removed this name and the others.

Results:

R: Throughout the results section, there is a recurring error in the abbreviation of Total Plasma Protein, which is incorrectly written as TTP. The correct abbreviation should be TPP

A: The text was rewritten with the change.

R: Page 8 line 17-18: “TTP and PCV showed the highest correlations overall, and using BCS, respectively, while FEC displayed the lowest correlations in both cases.” Something missing in the sentence and TTP should be TPP

A: It was a syntax issue, and the text was rewritten. It was indeed confusing.

Materials and Methods:

R: Page 24 line 15: Details on the number of pedigree records and the number of generations should be provided.

A: Details have been included.

R: Page 24 line 23: It is unclear what permanent environmental effects were tested in the model.

A: Permanent environmental effects were the fixed effects. The text was changed to clarify this, including its initial breakdown of generic and residual effects.

R: Page 25 line 7-12: The number of levels or classes for each fixed effect should be clearly specified.

A: The levels were discrete and an attempt was made to clarify the issue in the text regarding the unexplained effects.

R: Page 25 line 9: “age class (1 = 0 days, 2 = 1 to 150 days, 3 = 151 to 550 days, 4 = more than 550 days)” It is illogical to use a class for age zero days. How were measurements taken at zero age.

A: These data were tested in the context of a larger project, including birth weight. However, for the current study, this class has now been removed.

Tables and figures:

R: The quality of the figures is poor and they need to be reproduced in higher resolution.

A: The figures were generated again.

R: Table 1: The numbers for PCV are not presented as %

A: The text was rewritten with the change.

R: Table 1 and 2: Total Plasma Protein (TTP) should be Total Plasma Protein (TPP) in the header and inside tables

A: The text was rewritten with the change.

R: Table 2: I suggest to include genetic correlation

A: The text was rewritten with the change.

R: Table 3: “CV%” should be PCV %

A: The text was rewritten with the change.

R: Table 3: Standard deviations or standard errors should be added to the reported averages. Additionally, an ANOVA analysis comparing group means is missing and should be conducted to determine if the differences between groups are statistically significant.

Table S2 is not included and the link produce Table S1

A: We understand that this was unclear. The difference between groups in parasitology was significant at the genetic level and in the parameters analyzed, as confirmed by the analysis of variance and STRUCTURE. To assess the difference between farms, since the selective analysis contrasted the farm with the highest challenge (highest level of infection) and the farm with the lowest challenge (lowest level of infection), the results of a mean comparison test (also cited in the text) have now been added to Table 1, showing that all four traits differed.

---

## [Decision Letter · Decision Letter 2]

22 Sep 2025

Dear Dr. Menegatto,

We look forward to receiving your revised manuscript.

Kind regards,

Jean-Christophe Bambou

Academic Editor

PLOS ONE

Journal Requirements:

Reviewers' comments:

Reviewer's Responses to Questions

**Comments to the Author**

Reviewer #3: All comments have been addressed

2. Is the manuscript technically sound, and do the data support the conclusions?

Reviewer #3: Yes

3. Has the statistical analysis been performed appropriately and rigorously?

Reviewer #3: Yes

4. Have the authors made all data underlying the findings in their manuscript fully available?

Reviewer #3: Yes

5. Is the manuscript presented in an intelligible fashion and written in standard English?

Reviewer #3: Yes

Reviewer #3: The authors for the manuscript titled “Artificial and natural selection components reveal the mechanisms of tropical sheep populations against gastrointestinal parasites” have improved the manuscript and replied efficiently to the reviewer comments. The study and the manuscript provides good insights into understanding resistance and resilience to gastrointestinal nematodes. However, some points need minor changes.

General comments:

The table headers should be placed directly above the tables, with no text in between. Any accompanying text or explanation should follow the table as a separate paragraph, beginning with a reference to the table or figure number. For example, table 1 and 2 page 7.

Some paragraphs in the Results section and the beginning of the Discussion are too short and should be merged with the following paragraphs to improve flow and coherence.

Table 3. FEC units should be shift to same units as table 1, and standard deviations should be added.

Figure 3 and 4. The text within the figure should be enlarged in font size to ensure readability.

**Do you want your identity to be public for this peer review?** For information about this choice, including consent withdrawal, please see our Privacy Policy

Reviewer #3: **Yes:**  Hadeer M. Aboshady

---

## [Author Response · Author response to Decision Letter 3]

27 Nov 2025

# FIRST COMMENT

-- Due to editorial reasons related to copyright, Figure 1 was removed. Consequently, Figure 2 became Figure 1, and so on.

Figure 5 (in the new version, Figure 4) has been redone for the same reason.

# GENERAL COMMENTS

The table headers should be placed directly above the tables, with no text in between. Any accompanying text or explanation should follow the table as a separate paragraph, beginning with a reference to the table or figure number. For example, table 1 and 2 page 7.

A: The titles have been redone and we believe we have followed the recommendation now.

Some paragraphs in the Results section and the beginning of the Discussion are too short and should be merged with the following paragraphs to improve flow and coherence.

A: The paragraphs have been adjusted

Table 3. FEC units should be shift to same units as table 1, and standard deviations should be added.

A: The table was rebuilt with the correct FEC unit and standard deviations added. In fact, the FEC data was transformed and not readily intelligible.

Figure 3 and 4. The text within the figure should be enlarged in font size to ensure readability.

A: Figures 3 and 4 have been redone (in the new version, Figures 2 and 3).

---

## [Decision Letter · Decision Letter 3]

30 Dec 2025

Artificial and natural selection components reveal the mechanisms of tropical sheep populations against gastrointestinal parasites

PONE-D-24-55977R3

Dear Dr. Menegatto,

We’re pleased to inform you that your manuscript has been judged scientifically suitable for publication and will be formally accepted for publication once it meets all outstanding technical requirements.

Kind regards,

Jean-Christophe Bambou

Academic Editor

PLOS One

---

## [Editor Report · Acceptance letter]

PONE-D-24-55977R3

PLOS One

Dear Dr. Menegatto,

I'm pleased to inform you that your manuscript has been deemed suitable for publication in PLOS One. Congratulations! Your manuscript is now being handed over to our production team.

Kind regards,

on behalf of

Dr. Jean-Christophe Bambou

Academic Editor

PLOS One